# Surgical Management of Hemorrhoidal Disease in Inflammatory Bowel Disease: A Systematic Review with Proportional Meta-Analysis

**DOI:** 10.3390/jcm11030709

**Published:** 2022-01-28

**Authors:** Ugo Grossi, Gaetano Gallo, Gian Luca Di Tanna, Umberto Bracale, Mattia Ballo, Elisa Galasso, Andrea Kazemi Nava, Martino Zucchella, Francesco Cinetto, Marcello Rattazzi, Carla Felice, Giacomo Zanus

**Affiliations:** 1II Surgery Unit, Regional Hospital Treviso, AULSS2, 31100 Treviso, Italy; mattiaballo@gmail.com (M.B.); elisagalasso@gmail.com (E.G.); andreakazeminava@gmail.com (A.K.N.); martino.zucchella@aulss2.veneto.it (M.Z.); giacomo.zanus@aulss2.veneto.it (G.Z.); 2Department of Surgery, Oncology and Gastroenterology-DISCOG, University of Padua, 35128 Padua, Italy; 3Department of Medicine, Surgery and Neurosciences, Unit of General Surgery and Surgical Oncology, University of Siena, 53100 Siena, Italy; gaetanogallo1988@gmail.com; 4Statistics Division, The George Institute for Global Health, University of New South Wales, Sydney 2042, Australia; glditanna@gmail.com; 5Department of Advanced Biomedical Sciences, ‘Federico II’ University of Naples, 80131 Naples, Italy; umbertobracale@gmail.com; 6Department of Medicine-DIMED, University of Padua, 35128 Padua, Italy; francesco.cinetto@aulss2.veneto.it (F.C.); marcello.rattazzi@unipd.it (M.R.); carla.felice@aulss2.veneto.it (C.F.)

**Keywords:** hemorrhoidal disease, IBD, Crohn, ulcerative colitis, hemorrhoidectomy, surgery

## Abstract

Surgical treatment of hemorrhoidal disease (HD) in inflammatory bowel disease (IBD) has been considered to be potentially harmful, but the evidence for this is poor. Therefore, a systematic review of the literature was undertaken to reappraise the safety and effectiveness of surgical treatments in this special circumstance. A MEDLINE, Web of Science, Scopus, and Cochrane Library search was performed to retrieve studies reporting the outcomes of surgical treatment of HD in patients with Crohn’s disease (CD) and ulcerative colitis (UC). From a total of 2072 citations, 10 retrospective studies including 222 (range, 2–70) patients were identified. Of these, 119 (54%) had CD and 103 (46%) UC. Mean age was between 41 and 49 years (range 14–77). Most studies lacked information on the interval between surgery and the onset of complications. Operative treatments included open or closed hemorrhoidectomy (*n* = 156 patients (70%)), rubber band ligation (*n* = 39 (18%)), excision or incision of thrombosed hemorrhoid (*n* = 14 (6%)), and doppler-guided hemorrhoidal artery ligation (DG-HAL, *n* = 13 (6%)). In total, 23 patients developed a complication (pooled prevalence, 9%; (95%CI, 3–16%)), with a more than two-fold higher rate in patients with CD compared to UC (11% (5–16%) vs. 5% (0–13%), respectively). Despite the low quality evidence, surgical management of HD in IBD and particularly in CD patients who have failed nonoperative therapy should still be performed with caution and limited to inactive disease. Further studies should determine whether advantages in terms of safety and effectiveness with the use of non-excisional techniques (e.g., DG-HAL) can be obtained in this patient population.

## 1. Introduction

Hemorrhoids are clusters of smooth muscle, vascular, and connective tissue arranged in three columns along the anal canal, which contribute to continence mechanisms of healthy individuals. Hemorrhoidal disease (HD) refers to a pathologic or symptomatic process involving hemorrhoids and represents one of the most common problems leading patients to doctors all around the world, particularly in the Western countries, with a prevalence of 5–35% in the overall population [1,2].

Bleeding, anal swelling, prolapse, discomfort, pain, discharge, hygiene problems, and pruritus are the most common complaints [3]. The predominance or concomitance of one or more symptoms may reflect the large spectrum of pathological alterations of the internal or external hemorrhoids (i.e., laying above or below the dentate line, respectively).

Inflammatory bowel diseases (IBDs) are characterized by repetitive episodes of inflammation of the gastrointestinal tract caused by an abnormal immune response to gut microflora. IBDs encompass two idiopathic intestinal diseases: ulcerative colitis (UC) and Crohn’s disease (CD). While the former involves diffuse inflammation of the rectal and colonic mucosa, CD results in transmural ulceration that may ubiquitously affect the gastrointestinal tract [4]. In the Western world, the mean incidence for the period 2001–2014 soared to 20 per 100,000 [5].

To date, there is no consensus in the scientific literature regarding the exact indications for surgery in this special circumstance [6]. While some recommended that surgical procedures can only be considered in absence of active disease [7,8], others elected not to adopt a position [9]. Moreover, it remains uncertain whether more recently developed techniques may offer advantages over excisional hemorrhoidectomy.

The aim of this systematic review was to reappraise the safety and effectiveness of surgical treatments for HD in IBD.

## 2. Materials and Methods

The authors developed the protocol for review, in line with the Preferred Reporting Items for Systematic Reviews and Meta-Analyses (PRISMA) 2020 statement [10]. Although a review protocol was not registered prospectively, the primary objectives and methods were specified in advance.

### 2.1. Study Characteristics

Study characteristics were defined using the PICOS framework. Population: The review aimed to identify studies of patients diagnosed with IBD undergoing surgical interventions with the primary intent of treating HD. Intervention: Any surgical procedures for HD were included. Comparison: Studies were eligible regardless of whether they were prospective or retrospective in design, controlled or uncontrolled. Outcomes: Studies were broadly eligible if they provided extractable data on treatment efficacy, harm, or both.

All published reports to the date of final search (1 December 2021) were systematically reviewed.

### 2.2. Information Sources and Study Selection

The authors performed a comprehensive search of the literature using MEDLINE (PubMed), Embase, Cochrane Library of Systematic Review, Web of Science, Scopus, and by hand. Search term definitions were inclusive, promoting a sensitive search of studies reporting patients with IBD who underwent surgery for HD (Appendix A). Only full manuscripts and conference abstracts written in the English language (at least the abstract) were considered. Studies that exclusively reported on patients diagnosed with IBD after surgical treatment of HD were excluded as well as those describing the outcome of surgery limited only to removal of skin tags rather than treating HD, unless data on the selected population could be segregated from the total cohort. Studies were excluded if reporting on non-surgical treatments for HD (e.g., selective intra-arterial embolization) or if the outcomes of surgery in patients with IBD could not be segregated from the total study population. Reviews, guidelines, and editorials were also excluded.

### 2.3. Data Extraction

Screening was performed at the abstract level by two authors (M.B. and E.G.), excluding studies not meeting eligibility criteria where these could be readily determined from the abstract alone. Full-text copies of remaining studies were also obtained and assessed by the above authors, who were unblinded to the names of studies, authors, institutions, and year of publication. Disagreement regarding inclusion was resolved by a senior author (U.G.). Study characteristics and outcome data were extracted independently into a Microsoft Excel spreadsheet (XP professional edition; Microsoft Corp, Redmond, WA, USA), with disagreements resolved by consensus.

The following data were extracted for each study: publication year; study design; reason for exclusion; study period (in months); total number of subjects; gender; mean or median age and range; type of IBD; presence of other perianal disease; number of subjects in remission at surgery, on corticosteroids, immunomodulators, or biologics; type(s) of surgery; mean operative time; type of complications; mean follow-up (in months); recurrence; and satisfaction rate.

Qualitative assessment of studies was performed using the JBI Critical Appraisal Checklist for Case Series (https://synthesismanual.jbi.global, accessed on 1 January 2022). Two reviewers (M.B. and E.G.) independently performed the risk of bias evaluation and categorized the included articles as ‘high risk’ when the study bias rating ‘yes’ score was between 0% and 49%, ‘moderate risk’ when the study ‘yes’ score was between 50% and 69%, and ‘low risk’ when the study ‘yes’ score was above 70%. Any disagreement was resolved by consensus with a third author (U.G.).

### 2.4. Statistical Analysis

Meta-analyses of proportions of complications were performed using a random effects model with 95% confidence intervals (CIs) calculated using the Wilson method. Heterogeneity was assessed by a formal test of homogeneity and by the proportion of variability across studies attributable to heterogeneity rather than chance (I^2^). Meta-analyses were performed using the metaprop_one command in Stata 16 (StataCorp LLC, College Station, TX, USA).

## 3. Results

### 3.1. Selection of Sources of Evidence

After 175 duplicates were removed, a total of 2072 citations were identified from searches of electronic databases (*n* = 2061) and in-text citations (*n* = 11) (Figure 1).

Based on the title and the abstract, 2001 were excluded, with 71 full-text articles to be retrieved and assessed for eligibility. Of these, 61 were excluded, thus leaving 10 studies considered eligible for this review (Table 1). All were retrospective case series from five countries. There were two conference abstracts [11,12] and eight full-text articles [13,14,15,16,17,18,19,20]. Two (20%) studies were multicenter [18,19].

The 10 studies reported on a total of 222 patients, of whom 119 (54%) had CD and 103 (46%) UC (Appendix A). Mean age was between 41 and 49 years (range 14–77). Gender distribution was available in only six studies, with male predominance (54–71%) in all except one study (44%) [19]. Only the two most recent studies [18,19] reported the number of patients on corticosteroids, immunomodulators, and/or biologics at the time of surgery, who ranged between 17% and 25%. Only one study [20] characterized HD according to the Goligher classification [21]. The presence of other perianal disease (except for skin tags) affected one third of patients at the time of surgery in two studies [11,19]. Instead, patients with these characteristics were clearly excluded in one study [20].

### 3.2. Type of Surgery and Complications

Operative treatments included open or closed hemorrhoidectomy (*n* = 156 (70%), of whom 83 (53%) had CD and 73 (47%) UC), rubber band ligation (*n* = 39 (18%)), excision or incision of thrombosed hemorrhoid (*n* = 14 (6%)), and doppler-guided hemorrhoidal artery ligation (*n* = 13 (6%)) (Table 1).

Overall, 23 patients developed complications (pooled prevalence, 9% (95%CI, 3–16%); I^2^, 60.41%; Appendix A), with a more than two-fold higher rate in patients with CD compared to UC (11% (5–16%) vs. 5% (0–13%), respectively)

Anal abscess or fistula occurred in four (4.8%) patients with CD and one (1.4%) with UC, as opposed to urinary retention, which occurred more frequently in UC than CD (*n* = three (4.1%) vs. one (1.2%) patient, respectively). After OEH/CEH, anal stenosis (*n* = 3 (3.6%)), bleeding (*n* = 2 (2.4%)), non-healing wounds (*n* = 2 (2.4%)), and anal fissure (*n* = 1 (1.2%)) were only observed in patients with CD. Two (2.7%) patients with UC suffered uncontrolled anal pain. The overall morbidity rate after OEH/CEH in IBD was 12% (19/156 patients).

Out of 39 patients undergoing RBL (19 (49%) with CD and 20 (51%) with UC), one (2.6%) case of anal stenosis was observed in a patient with CD.

There were no cases of major immediate postoperative complications after DG-HAL, but 3/13 (23%) patients with CD suffered from recurrent hemorrhoidal bleeding during the 18-month follow-up period.

### 3.3. Risk of Bias within Studies

Except the two most recent publications [18,19], all studies presented moderate to high risk of bias. The question which most commonly elevated the risk of bias was ‘Was there clear reporting of clinical information of the participants?’ (Appendix A).

## 4. Discussion

This systematic review showed a 9% (95%CI, 3–16%) pooled rate of complications after surgical treatment of HD in IBD, with a more than two-fold higher rate in patients with CD as compared to UC (11% (5–16%) vs. 5% (0–13%)), respectively.

In a large retrospective study of 50,000 treated subjects with HD, Jeffery et al. [13]. demonstrated unacceptable morbidity in IBD patients, affecting two thirds of those with CD. Subsequent series confirmed these findings, thus dampening through time surgical inclination for HD in IBD.

It was originally suggested that symptomatic hemorrhoids rarely occur in patients with CD and that in many cases skin tags are really being treated instead of hemorrhoidectomy [22]. Furthermore, several series included both patient groups (i.e., those undergoing surgical management of HD or excision of skin tags) [15,17,18,23]. One of the strengths of our review is the exclusion of patients undergoing excision of skin tags rather than surgical treatment of HD. Skin tags affect 11–19% of patients with CD [24,25] and are rarely symptomatic. Conservative management has long been advocated in light of an increased risk of delayed wound healing and exacerbation of pre-existing perianal CD if excised [15,26]. The inclusion of mixed populations in some studies may have overestimated the risk of complications, especially in CD [23].

The risk of complications in patients with CD was slightly lower than that observed in a previous systematic review of 99 patients (14% vs. 17%, respectively) [23] and likely reflects our choice to exclude patients treated for HD with a diagnosis of IBD made post-operatively. Indeed, two studies showed a significantly higher prevalence of complications in this group of patients, compared to those with a known diagnosis at the time of surgery [13,17]. It is possible that individuals already diagnosed with IBD may have been less likely to undergo more aggressive treatment (e.g., ≥3-quadrant OEH/CEH).

Although the traditional excisional methods (OEH and CEH) still remain the gold standard for symptomatic III- and IV-degree HD [6], a large systematic review and network meta-analysis showed that the risk of complications after DG-HAL is 38% and 59% lower than OEH and CEH, respectively [27]. Even if based on a single center series, DG-HAL appears a safer technique in IBD.

Despite focusing on the surgical management of HD by excluding other forms of perianal surgery in IBD (e.g., excision of skin tags), this systematic review has several limitations: first, the overall poor quality of included studies, all being small retrospective and uncontrolled series from single centers, with moderate to high risk of bias. Second, studies were more than 40 years apart, with various uncontrolled factors that may have influenced our findings (e.g., inter-surgeon procedural variations; allied procedures, e.g., for anal fissure concomitant to HD; patient selection bias; use of medications; IBD activity at the time of surgery; unclear time to complication in most cases).

## 5. Conclusions

Despite these caveats, the results of this review may help raise awareness in the surgical community of the importance of the topic and serve to promote future studies aimed at shedding more light on the safety of traditional and newer techniques to treat HD in this special circumstance.

## Figures and Tables

**Figure 1 jcm-11-00709-f001:**
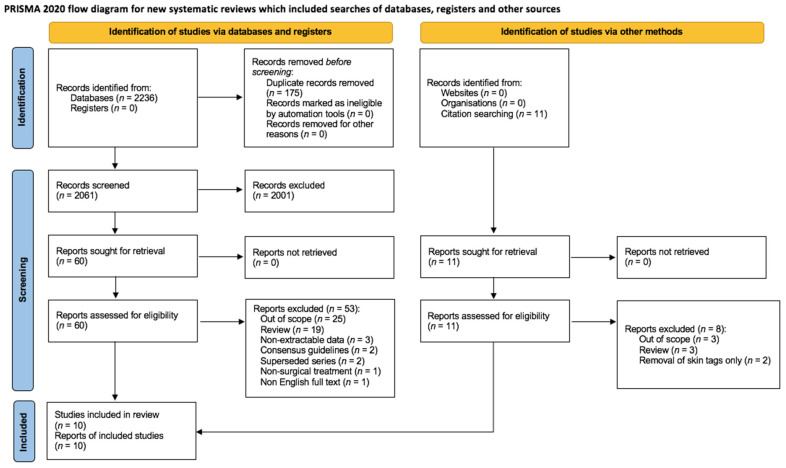
PRISMA 2020 flow diagram.

**Table 1 jcm-11-00709-t001:** Characteristics of the retrospective case series.

First Author	Year	Country	Study Period (Years)	N *	CD	UC	Active IBD at Surgery	Type of Operation	Complications (N)		Average Follow-Up (Months)
									CD	UC	
Jeffery [13]	1977	UK	40	20	4	16	nr	OEH	1	3	480
Hughes [14]	1978	UK	6	2	2	0	nr	OEH	1	/	72
Keighley [15]	1986	UK	1	2	2	0	nr	RBL	1	/	12
Wolkomir [16]	1993	USA	15	17	17	0	0	CEH ^	4	/	137
Karin [20]	2012	Israel	4	13	13	0	nr	DGHAL	3	/	18
D’Ugo [17]	2013	Italy	8	9	9	0	0	Mixed §	1	/	37
Koh [11]	2015	USA	14	9	9	0	1	OEH	0	/	28
Lee [12]	2017	S. Korea	11	44	0	44	nr	OEH	/	0	nr
McKenna [18]	2019	USA °	17	70	27	43	nr	Mixed ¶	2	3	29
Lightner [19]	2020	USA °	24	36	36	0	nr	OEH	4	/	31.5

IBD: inflammatory bowel disease; CD: Crohn’s disease; UC: ulcerative colitis; OEH: open excisional hemorrhoidectomy; RBL: rubber band ligation; CEH: closed excisional hemorrhoidectomy; DGHAL: doppler-guided hemorrhoidal artery ligation; nr: not reported. * Total number of patients already diagnosed with IBD at the time of surgery for HD, which may not correspond to the total number of study patients. ^ Less than 3-quadrant in 5/17 (29%) patients. ° Multicenter. § Included OEH (*n* = 6), CEH (*n* = 1), RBL (*n* = 2). ¶ Included RBL (*n* = 35), OEH or CEH (*n* = 21), excision or incision of thrombosed hemorrhoid (*n* = 14).

## Data Availability

The data presented in this study are available on request from the corresponding author.

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
