# Peer review of "Surgical Management of Hemorrhoidal Disease in Inflammatory Bowel Disease: A Systematic Review with Proportional Meta-Analysis"

_jcm, 2022, doi:10.3390/jcm11030709_

Round 1

Reviewer 1 Report

This article is to review the evidence of the surgical treatments for hemorrhoids in patients with IBD. I have several comments:

  1. In the abstract, the authors need to state their study aim and the results more clearly. Is the outcome good or bad for surgical treatments? Is the evidence convincing?
  2. In the methods section, the study characteristics included the selection criteria, which should be described in detail according to the PICOS principle. 
  3. The meta-analysis showed an I2 of 99.6%, which suggests that the meta-analysis might not be performed. Otherwise, you might have to explore the source of heterogeneity.
  4. I think the Tables and Figures should not be embedded in the article, especially supplementary table and figure were embedded.

Author Response

We thank the 2 reviewers for their comments. We have responded to all of these in turn below with edits in the revised manuscript indicated by yellow font as per journal instructions.

RESPONSE TO REVIEWERS’ COMMENTS

Reviewer #1:

This article is to review the evidence of the surgical treatments for hemorrhoids in patients with IBD. I have several comments:

In the abstract, the authors need to state their study aim and the results more clearly. Is the outcome good or bad for surgical treatments? Is the evidence convincing?

We thank the reviewer for his/her comment. We have edited the abstract accordingly.

In the methods section, the study characteristics included the selection criteria, which should be described in detail according to the PICOS principle.

We have edited this section as follows:

2.1 Study characteristics

Study characteristics were defined using the PICOS framework. Population: the review aimed to identify studies of patients diagnosed with IBD undergoing surgical interventions with the primary intent of treating HD. Intervention: any surgical procedures for HD. Comparison: studies were eligible regardless of whether they were prospective or retrospective in design, controlled or uncontrolled. Outcomes: studies were broadly eligible if they provided extractable data on treatment efficacy, harms or both.

The meta-analysis showed an I2 of 99.6%, which suggests that the meta-analysis might not be performed. Otherwise, you might have to explore the source of heterogeneity.

We thank the reviewer for his/her comment. Indeed, we are aware of different sources of heterogeneity across the studies included in our systematic review. We deemed appropriate to offer some insights to the readers on of the pooled proportions of complications. It is quite expected by the way it is calculated (which is the proportion of variability attributable to heterogeneity rather than by sampling error) in proportional meta-analysis, to have quite high values of I2. Our pooled proportion and its 95% confidence interval is mainly exploratory. We have made this clearer by slightly amending the Title (e.g. Surgical management of hemorrhoidal disease in inflammatory bowel disease: a systematic review with proportional meta-analysis) and the Methods section as follows:

2.4 Statistical analysis

Meta-analyses of proportions of complications were performed using a random effects model with 95% confidence intervals (CI) calculated by the Wilson method.

I think the Tables and Figures should not be embedded in the article, especially supplementary table and figure were embedded.

We thank the reviewer for his/her comment. We have left only Table 1 in the manuscript and moved the other Figures/Tables to the Supplements.

Reviewer 2 Report

Grossi et al. conducted a systematic review and meta-analysis about surgical management of hemorrhoidal disease in inflammatory bowel disease.

Although the topic is interesting, this reviewer has several concerns as follows.

1) This reviewer is not sure how they conducted a meta-analysis. What ES means? The authors should spell out this abbreviation in the Figure legend. What measure (odds ratio or something else for what?) did they pool? The author should mention this in the method section. It is weird to see some numbers [e.g., ES 1.00 (95% CI, 0.70-1.00), 0.00 (95% CI, 0.00-0.08)]. Usually, point estimates are between upper and lower 95% CIs. I wonder if these numbers are not correct. Also, there is no description for this meta-analysis. How did they interpret the result of this meta-analysis? The author should add such description (including study heterogeneity) in the result section.

2) The study design can be added in Table 1. If possible, it would be better not to use many abbreviations. That would reduce readability.

3) The date when the author conducted literature review (14 April, 2021) was old. The author should update the systematic review.

4) Is there any similar meta-analysis or systematic review? If there is any, the authors should cite those papers. If not, the authors should mention that there is no prior meta-analysis or systematic review on this topic.

Author Response

We thank the 2 reviewers for their comments. We have responded to all of these in turn below with edits in the revised manuscript indicated by yellow font as per journal instructions.

RESPONSE TO REVIEWERS’ COMMENTS

Reviewer #2:

Grossi et al. conducted a systematic review and meta-analysis about surgical management of hemorrhoidal disease in inflammatory bowel disease.

Although the topic is interesting, this reviewer has several concerns as follows.

1) This reviewer is not sure how they conducted a meta-analysis. What ES means? The authors should spell out this abbreviation in the Figure legend.

We thank the Reviewer for his/her comment. We have replaced the acronym ES (i.e. effect size) with ‘proportions’ in the Forest plot.

What measure (odds ratio or something else for what?) did they pool? The author should mention this in the method section. It is weird to see some numbers [e.g., ES 1.00 (95% CI, 0.70-1.00), 0.00 (95% CI, 0.00-0.08)]. Usually, point estimates are between upper and lower 95% CIs. I wonder if these numbers are not correct. Also, there is no description for this meta-analysis. How did they interpret the result of this meta-analysis? The author should add such description (including study heterogeneity) in the result section.

We thank the reviewer for his/her supportive comments. Indeed, only meta-analysis of proportions of complications could be performed. We have made this clearer by slightly amending the Title and the Methods section. Please see response to Reviewer #1, point #3 as follow:

The meta-analysis showed an I2 of 99.6%, which suggests that the
meta-analysis might not be performed. Otherwise, you might have to
explore the source of heterogeneity.

We thank the reviewer for his/her comment. Indeed, we are aware of
different sources of heterogeneity across the studies included in our
systematic review. We deemed appropriate to offer some insights to the
readers on of the pooled proportions of complications. It is quite
expected by the way it is calculated (which is the proportion of
variability attributable to heterogeneity rather than by sampling error)
in proportional meta-analysis, to have quite high values of I2. Our
pooled proportion and its 95% confidence interval is mainly exploratory.
We have made this clearer by slightly amending the Title (e.g. Surgical
management of hemorrhoidal disease in inflammatory bowel disease: a
systematic review with proportional meta-analysis) and the Methods
section as follows:

2.4 Statistical analysis
Meta-analyses of proportions of complications were performed using a
random effects model with 95% confidence intervals (CI) calculated by
the Wilson method.

2) The study design can be added in Table 1. If possible, it would be better not to use many abbreviations. That would reduce readability.

We thank the reviewer for his/her supportive comment. We did originally plan to include the study design. This was later deleted as all studies were retrospective series. We have added this information in the title of the Table as follows:

Table 1. Characteristics of the retrospective case series

3) The date when the author conducted literature review (14 April, 2021) was old. The author should update the systematic review.

 We thank the reviewer for his/her supportive comments. We have extended the search to Dec 1st 2021 with no additional yield, and edited the text as follows:

All published reports to the date of final search (01 December 2021) were systematically reviewed.

4) Is there any similar meta-analysis or systematic review? If there is any, the authors should cite those papers. If not, the authors should mention that there is no prior meta-analysis or systematic review on this topic.

We thank the reviewer for his/her supportive comment. We have discussed the strengths of our systematic review as compared to the most recently published one in 2014. Indeed, in the Discussion we stated “The risk of complications in patients with CD was slightly lower than that observed in a previous systematic review of 99 patients (14% vs. 17%, respectively),[23] and likely reflects our choice to exclude patients treated for HD with a diagnosis of IBD made post-operatively. Indeed, two studies showed a significantly higher prevalence of com-plications in this group of patients, compared to those with a known diagnosis at the time of surgery.[13, 17] It is possible that individuals already diagnosed with IBD may have been less likely to undergo more aggressive treatment (e.g. ≥3-quadrant OEH/CEH).”

Round 2

Reviewer 2 Report

This reviewer is not still sure how the authors calculated proportions of complications in each study. For example, in Hughes et al. the number of complication is 1 and the total case number is 2, so proportion should be 0.5, but why it is 1.0 in the meta-analysis Figure? In addition, in Koh et al, the number of complications is 0, but in the meta-analysis figure, the proportion is 1.0. The authors need to fix (i.e., add the numerators and denominators for proportions of complication) or explain this discrepancy. 

Author Response

This reviewer is not still sure how the authors calculated proportions of complications in each study. For example, in Hughes et al. the number of complication is 1 and the total case number is 2, so proportion should be 0.5, but why it is 1.0 in the meta-analysis Figure? In addition, in Koh et al, the number of complications is 0, but in the meta-analysis figure, the proportion is 1.0. The authors need to fix (i.e., add the numerators and denominators for proportions of complication) or explain this discrepancy.

We thank the reviewer for spotting this discrepancy. Indeed, Figure S1 actually showed the proportion of No. patients with Crohn’s disease / Total IBD Patients. Studies included in the proportional meta-analysis of complications (i.e. No. complicated / Total Patients) have now been captured in Figure S2. We have amended the legend accordingly and renumbered these Figures throughout the text.